# The performance of interrupted time series designs with a limited number of time points: Learning losses due to school closures during the COVID-19 pandemic

**Florian D. van Leeuwen** [1] *, **Peter Lugtig**[1], **Remco Feskens**[2]

**1** Department of Methods and Statistics, Faculty of Social Science, Utrecht University, Utrecht, The Netherlands, **2** Cito, Arnhem, The Netherlands

* f.dammesvanleeuwen@gmail.com

**Data Availability Statement:** The data underlying the results presented in the study are available from CITO (https://cito.nl/over-cito/contact/).

## Abstract

Interrupted time series (ITS) designs are increasingly used for estimating the effect of shocks in natural experiments. Currently, ITS designs are often used in scenarios with many time points and simple data structures. This research investigates the performance of ITS designs when the number of time points is limited and with complex data structures. Using a Monte Carlo simulation study, we empirically derive the performance—in terms of power, bias and precision- of the ITS design. Scenarios are considered with multiple interventions, a low number of time points and different effect sizes based on a motivating example of the learning loss due to COVID school closures. The results of the simulation study show the power of the step change depends mostly on the sample size, while the power of the slope change depends on the number of time points. In the basic scenario, with both a step and a slope change and an effect size of 30% of the pre-intervention slope, the required sample size for detecting a step change is 1,100 with a minimum of twelve time points. For detecting a slope change the required sample size decreases to 500 with eight time points. To decide if there is enough power researchers should inspect their data, hypothesize about effect sizes and consider an appropriate model before applying an ITS design to their research. This paper contributes to the field of methodology in two ways. Firstly, the motivation example showcases the difficulty of employing ITS designs in cases which do not adhere to a single intervention. Secondly, models are proposed for more difficult ITS designs and their performance is tested.

## Introduction

The randomized controlled trial (RCT) is regarded as the gold standard for evaluating the effectiveness of interventions as it allows one to ascertain the internal validity of causal claims [1]. In a RCT, a researcher introduces an intervention and participants are randomly assigned to one or more treatments [2]. For some interventions, such as policy changes, random assignment is not possible. Natural experiments are often used instead, where researchers have no

**Funding:** The author(s) received no specific funding for this work.

**Competing interests:** The authors have declared that no competing interests exist.

influence over when the intervention happens or who is assigned to the treatment and control groups [3].

Natural experiments are extremely important for policymakers, often being the only way to estimate the effect of a policy change. The results of such studies influence the implementation of new policies. Examples of natural experiments include: what is the effect of school closures during the COVID-19 outbreak on students' learning [4], do physical education requirements reduce youth obesity [5], or what is the impact of measures to restrict access to commonly used means of suicide [6]? The researcher tries to answer a question without participating in the process of allocation of treatments. In these cases, the entire population might be in the treatment group or the treatment group could consist of a non-random sample. Data are derived from already existing administrative records or surveys, depriving the researcher of control of the sample size [7].

An option to estimate the effect of an intervention without having control over allocation is to make use of longitudinal data. There are multiple ways to estimate the effects of an intervention with longitudinal data [8–11]. Difference-in-difference (DD) is the most simple method. As explained by Goodman et al. [12]: "A DD estimate is the difference between the change in outcomes before and after a treatment (difference one) in a treatment versus control group (difference two)." The analysis only allows us to estimate a change in means, a gradual change over time cannot be modelled in a DD [13]. There exist, however, cases with an initial drop and a catch-up effect afterwards, resulting in a deceivingly small difference in means. The two effects with different signs are then crossed out against each other. Furthermore, researchers can be interested in dissecting the total effect into an initial shock and a change over time. In such cases, an alternative method to estimate is the interrupted time series (ITS) design. It is considered one of the best designs for establishing causality when RCTs are not possible [14]. The ITS design has the advantage that it allows us to model a general trend in the data while also detecting immediate (step) and gradual (slope) changes after an event. In an ITS design the trend before an intervention is extrapolated and compared to the observed trend after the intervention. The difference in these trends is considered as the effect of the intervention [15]. This assumes that all change within that timeframe is due to the intervention as the ITS is not able to exclude confounding other events occurring at a similar time as the interventions [16].

An ITS requires at least two time points before and after the intervention to model four parameters: an intercept, general slope, step change and slope change. The rule of thumb practice on how many time points before and after an intervention are necessary for an ITS design varies. Fretheim et al. [17] considered six or more as the minimum threshold for reliable results, Penfold et al. [18] concluded that at least eight are necessary, while Wagner et al. [19] recommend twelve data points before and after the intervention. In some cases the minimum is easily reached, Turner et al. [20] found a median of 48 time points in a sample of 200 ITS studies in PubMed. Similarly, a study into 116 ITS analyses in MEDLINE found the median number of time points to be 39 [21]. In other areas, such data may not be as abundant. For example, educational testing schemes usually have only 1–3 exams per year. In these settings, there are concerns about the power of an ITS design, potentially leading to the publication of weak and spurious findings [22, 23].

There have been some simulation studies looking into the influence of the number of time points, sample size, effect size, and autocorrelation on the statistical power and bias [20, 24, 25]. These studies show that increasing the number of time points, sample size or effect size results in an increase of power. The scenarios investigated were optimal by for example, only including a single step or slope change. Real-world situations are often much more complicated e.g., the general trend might change over time which could bias ITS estimates. Moreover, a policy change might be introduced and then tweaked not long after the initial intervention

[19, 26, 27]. Such a secondary intervention could again bias the ITS estimates if the model is not correctly specified. It is unclear how well the ITS design can perform in such cases.

This study addresses this gap in the literature by estimating how well the ITS design performs in scenarios with multiple interventions, few time points and small effect sizes. We aim to assess how well the ITS design performs under different, more complicated scenarios. In this research, such scenarios pertain to learning losses due to school closures during the COVID pandemic: there are usually only a limited number of measurements available (outcomes of standardized educational tests) and schools were closed in two periods to name two potential difficulties in the estimation process.

This paper aims to empirically derive the power, bias and precision for the ITS design in realistic scenarios. The remainder of this paper starts by discussing the classical ITS design in general terms and extensions that are made to accompany different situations. A motivating example of estimating learning losses due to COVID lockdowns using an ITS design is outlined. Then the methods of the simulation study to assess the performance of the ITS designs are given. Next, the results are presented. Finally, the findings are summarized and future research areas are outlined.

## Interrupted time series design

This section consists of two parts that explain how the ITS design can be implemented. Firstly, regression-based models for estimating ITS designs are discussed. Secondly, these models are extended by multilevel (ML), structural equation (SEM) and autocorrelation (AR) models.

### Linear regression models

In linear regression, an outcome (Y) is estimated by a predictor (X). This relationship can be expressed as a linear line such as in the equation below:

$$Y_i = \beta_0 + \beta_1 X + \epsilon_i, \tag{1}$$

where $\beta_0$ is the intercept (the value of $Y$ where $X$ is equal to zero), $\beta_1$ is the slope of the line and $\epsilon_i$ the error term. The most common way to obtain the estimates in the model is through ordinary least squares (OLS) estimation. The specifications about OLS and its assumptions can be found in Weisberg et al. [28].

**Segmented regression.**   Segmented regression is the most frequently used method for estimating ITS designs [21]. It divides the time series into parts based on the number of interventions. In each part, a regression line can be fitted with a free intercept and slope. A minimum of three parameters are required for an ITS analysis using a segmented regression:

- $T$ (an indicator of time)

- $X_t$ (a dummy (0/1) indicating the pre/post-intervention period); Information about the timing of the intervention is necessary.

- $Y_t$ (outcome at time point t)

The variable time should be centred around the time of the intervention. To estimate the effect of the intervention, the following OLS segmented regression model can be used [29]:

$$Y_t = \beta_0 + \beta_1 T + \beta_2 X_t + \beta_3 T X_t + \epsilon_t, \tag{2}$$

where $\beta_0$ indicates the baseline level at $T = 0$, $\beta_1$ represents the trend before the intervention, $\beta_2$ is the step change due to the intervention at $T = 0$ and $\beta_3$ is the slope change due to the intervention. There need to be at least two measurements of $Y_t$ before and after the intervention to

estimate the baseline slope ($\beta_1$) and the change in slope ($\beta_3$) since a slope can only be estimated if there is more than one data point.

This model can be extended in the case of multiple interventions by [30]:

$$Y_t = \beta_0 + \beta_1 T + \beta_2 X_{1t} + \beta_3 TX_{1t} + \beta_4 X_{2t} + \beta_5 TX_{2t} + \epsilon_t, \tag{3}$$

where $X_1$ is the first intervention and $X_2$ the second. $\beta_4$ indicates the total change in intercept since the second intervention. The step change of the second intervention can be expressed as $\beta_4 - \beta_2$. $\beta_5$ expresses the change in slope with regard to the pre-intervention slope $\beta_1$. Again there need to be at least two measurements between and after the two interventions to estimate the slope change in that time frame.

Another extension that can be made if the slope is expected to change over time, e.g., children learn less as they grow older. In that case, another parameter can be added to Eq 2:

$$Y_t = \beta_0 + \beta_1 T + \beta_2 X_t + \beta_3 TX_t + \beta_4 T^2 + \epsilon_t, \tag{4}$$

where $\beta_4$ indicates the change of slope over time. An interaction term may be introduced between $X_t$ and $T^2$ if the change of slope is expected to change after an intervention.

In a case with multiple interventions and where the slope is expected to change over time, the models in Eqs 3 and 4 can be combined into the following model:

$$Y_t = \beta_0 + \beta_1 T + \beta_2 X_{1t} + \beta_3 TX_{1t} + \beta_4 X_{2t} + \beta_5 TX_{2t} + \beta_6 T^2 + \epsilon_t. \tag{5}$$

## Extensions of linear regression models

There are multiple extensions of segmented regression beyond OLS models. ML models can be used, which can be further extended by the use of SEM models. Another extension of the OLS model is the use of models that correct for autocorrelation. These extensions will not be used in the simulation study but are included to give an overview of the possible models that can be used in the ITS design.

**Multilevel models.**   If the data stem not from a single population but from subpopulations, such as schools or clinics, multilevel models can be used to account for the clustering of observations [31]. The interest of an ITS analysis can be the difference in intercept or slope after an intervention, but the variation in such an effect is also an important factor. For example, do all people have the same response to the intervention or is there large variability? These growth curves can be modelled with a ML model [32]. Furthermore, using ML models has the advantage that individuals with a different number of time points or time points that are not equally spaced can still be estimated [32].

**Structural equation models.**   Another way to model growth over time is through growth curve models (GCM). In GCM models, a latent variable (e.g., math ability) is estimated by observed variables e.g., one variable measured at different time points. The focus in these models is on within-person change [33]. An ITS design in a SEM context is known as a piecewise growth curve model (PGCM) [34]. The term originates from piecewise regression which is identical to segmented regression. PGCMs are capable of detecting simultaneous differences in both intercept and slope and provide tests of significance for these two necessary indicators of an ITS intervention effect [35]. The advantage of modelling through SEM compared to ML is that extensions are easier to make [33]. For example, if we think that the time points before the intervention might be more correlated then we can add another term to the PGCM to let the errors correlate [36].

**Autocorrelation models.**   An often cited downside to using OLS models in an ITS design is that it does not take into account the effect of autocorrelation [25, 37]. A model that does

take this into account is the autoregressive (AR) model [38]. In an AR model, the outcome variable is predicted by one or multiple lagged values. An AR-1 model, with one lagged value, is often used in ITS designs [39].

An extension of the AR model that is used in ITS designs is the autoregressive integrated moving average (ARIMA) model. The ARIMA model consists of three parts: an AR model, differencing and a moving average (MA) model [37]. The AR part consists of lagged Y values. Differencing is used to achieve stationarity by taking the difference of the outcome variable between two time points [38]. The MA part is a lagged error term.

The ARIMA model can be further extended by controlling for seasonality, that is a variation occurring at regular intervals. In educational data, seasonality can for example be due to school holidays. In ARIMA modelling, a way to deal with seasonality is to take the seasonal difference [40]. Although the ARIMA model can handle autocorrelation well, it might not be the most practical model in many natural experiments. The literature often states that the ARIMA model needs at least 50 time points [41], but preferably 100 or more [42].

This study is interested in population estimates, not in within-person change. For this reason, the ML and SEM models are not used in the simulation study. Moreover, Turner et al. [20] demonstrate in a simulation study into statistical methods used for ITS designs that OLS models outperformed models that correct for autocorrelation when the number of time points was lower than twelve. As the focus of this study is data with few time points, OLS models are used for the simulation study.

## Motivating example

During the COVID pandemic lockdowns, many (primary) schools changed from on-location education to an online variant of education. Numerous studies have looked into how the shift in education methods has impacted children's learning [4, 43–45]. The objective of these studies is to estimate the average (causal) effect of school closures on children's learning. Only a limited number of time points are used in these studies as students usually do not take more than one standardised test per year.

In this study, an ITS design is used to estimate the effects of the lockdowns. Different ITS designs are applied to show the stability of models with more parameters when the number of time points is limited.

## Data

The dataset that is used in the motivating example consists of educational test scores of primary schools for pupils in grades 4–7 in the Netherlands. The test scores obtained on different test versions in different years are comparable through item response theory (IRT) models [46]. About 80% of primary schools participate in this student monitoring system which can be used to track the learning progress of pupils. Each participating school tests the pupils in January and May every year. The test consists of multiple parts, such as reading comprehension, math and spelling. This study focuses only on the mathematical part as most data were available for this subject. The dataset has a longitudinal cohort nature [47], each cohort is measured at the same time point in their school career but not at the same point in time. This study will focus on cohorts that have at least two observations before and after the lockdowns so that a change in slope can be estimated. The data available cohorts can be seen in Table 1.

The study, 22–1805, has received written approval by the Ethical Review Board of the Faculty of Social and Behavioural Sciences of Utrecht University. The need for consent was waived by the Ethical Review Board. The fully anonymous, data was accessed on 11–10-2022.

**Table 1. Longitudinal cohort data.**

|          | 2017 | 2018 | 2019 | 2020 | 2021 | 2022 |
|----------|------|------|------|------|------|------|
| Cohort A |      | O    | O    | O    | O    |      |
| Cohort B |      |      | Z    | Z    | Z    | Z    |

O indicates the presence of data for cohort A, similarly for Z with cohort B.

There are 8 observations per student and between 30,000–50,000 students per time point, after filtering duplicates and missings. There is a hierarchical structure of time points within students and students within schools [31]. Fig 1 shows the differences in slopes for a sample of students from different cohorts. The starting ability and learning rate are not the same for all students, which is in line with the literature [48]. The grey area indicates the two school closures due to COVID. The first lockdown was approximately two and a half months starting in the middle of March 2020. The second lockdown was shorter, only a month and a half starting in the middle of December 2020.

## Results for the motivating example

In the analysis ML models, correcting for the hierarchical structure of the data, are used to estimate different ITS designs. A random intercept model was used, meaning each student obtained an individual intercept. Most variation was observed in the intercepts, adding a random slope of time did not converge due to the homogeneity of the slopes. Time is centred around the start of the first lockdown. The model equations used were similar to those of the segmented regression section with the addition of random intercepts.

The effects of the two lockdowns were modelled with several different models. The first is the basic segmented regression (Eq 2), where the effect of the two lockdowns is modelled into a single parameter for the step change ($\beta_2$). This is followed by modelling the step change for the second intervention separately (Eq 3). Since there is only one time point between the interventions, it is not possible to separate the slope change between the interventions. The term in Eq 3 regarding the separate slope change for after the second intervention $\beta_5 TX_{2t}$ is thus not included. In the third model (Eq 4) time squared is added as a parameter, to account for the fact that students learn more in their first years than in the later part of their career [49]. The last model consists of a combination of the second and third models (Eq 5).

In Table 2 Model 1a is the baseline ITS design for cohort A. The intercept is 233.45 and indicates the mean math score of all students at the start of the first lockdown. The pre-intervention slope is 29.46, the step change is -5.22 and the slope change is -7.34. In Model 2a we see that the step change for the second intercept is 0.89 ($\beta_{COVIDlockdown} - \beta_{SecondCOVIDlockdown} = -5.47 - -6.36 = 0.89.$) and the estimate for slope change is unstable, as the estimates deviate largely from the estimates in the baseline model. In Model 3A time squared is added to the model and all estimates turn unstable. Model 4A is again very unstable. A possible reason for this instability is that there are only three time points after the first intervention, and three parameters are trying to be estimated into that segment.

In Table 3 the effects of the COVID lockdown are shown for cohort B. The main differences are that models 2B and 3B are more stable, as the estimates stay relatively close to the baseline model. The coefficient for the change in slope is around the same value compared to the baseline model. Introducing the extra terms for the second intervention and time squared does not lead to large changes in the parameters. Model 4B, which has both extra parameters, is again very unstable as can be seen in the decrease of the slope parameter and increase of the change in slope parameter.

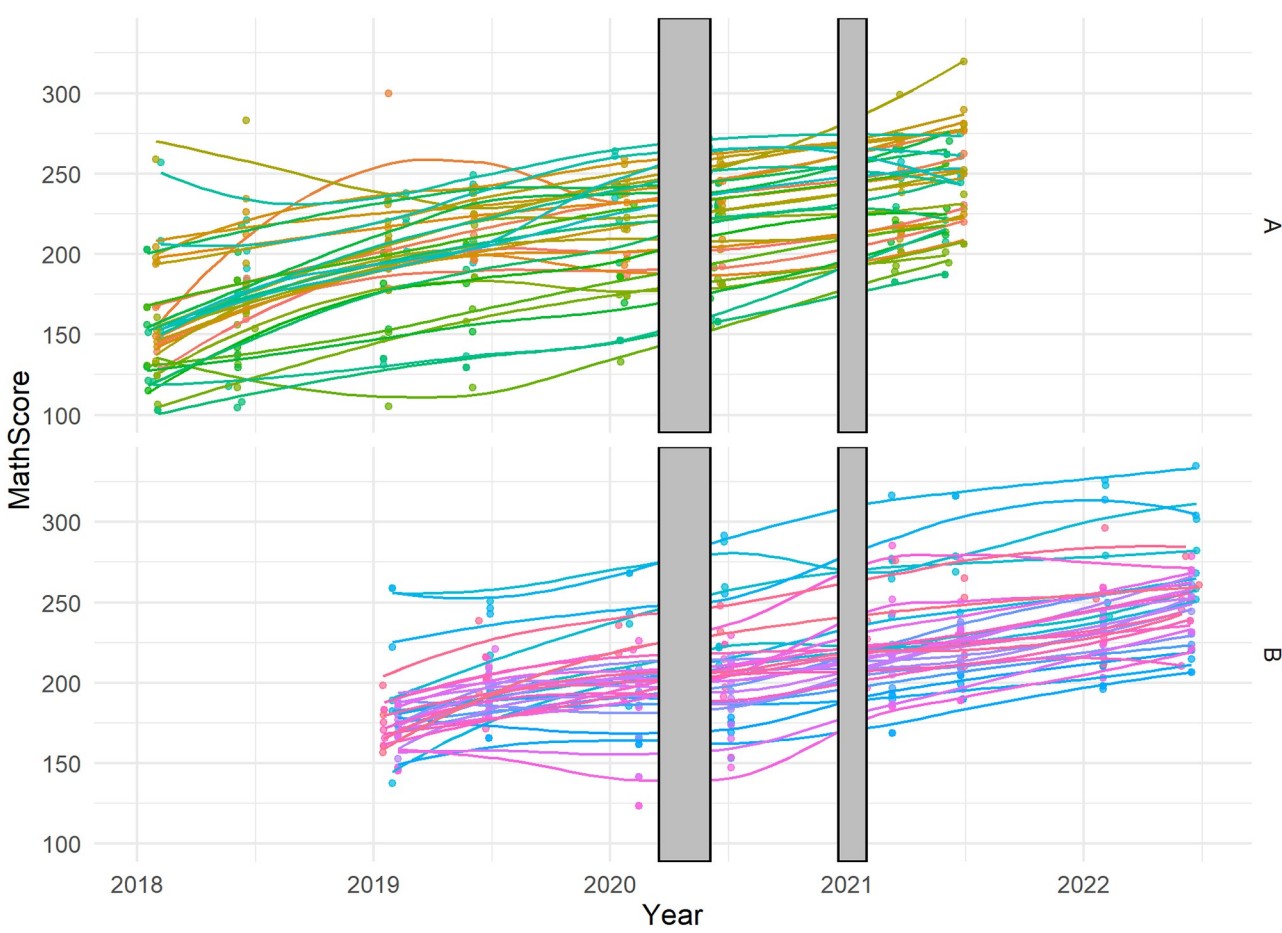

**Fig 1. Individual growth curves for 25 students per cohort sampled randomly from CITO data.** The grey area represents the lockdown periods.

From the output, it seems that it is difficult to estimate extended versions of the ITS design with only eight time points, especially when the intervention occurs late in the time series. The variation in the estimates might be due to overfitting [50]. A simple way to combat overfitting is to include fewer parameters but this will then be an underfit to the data. To make an informed decision on which model to use, it is essential to understand how robust the ITS design is against over- and underfitting. To investigate this relation a simulation study will be used.

## Methods

A Monte Carlo simulation study is used to assess the performance of different ITS designs over 384 scenarios. Monte Carlo simulation allows us to estimate our statistics of interest by eliminating the noise caused by random sampling [51]. This allows us to inspect under which circumstances the ITS designs can estimate an effect well. In the simulation study, data are generated based on predefined parameters. These are then compared to the estimates to evaluate the performance of the models.

### ITS scenarios

Data are simulated to reflect the motivating example. Table 4 presents the data-generating parameters for 384 data-generating scenarios. For each data-generating scenario, 1,000

**Table 2. Results of different ML models with an ITS design for cohort A.**

|  | Model 1A | Model 2A | Model 3A | Model 4A |
|---|---|---|---|---|
| (Intercept) | 233.45*** | 233.45*** | 226.47*** | 226.21*** |
|  | (0.15) | (0.15) | (0.16) | (0.16) |
| Time | 29.46*** | 29.46*** | 9.84*** | 9.11*** |
|  | (0.04) | (0.04) | (0.16) | (0.16) |
| COVID lockdown | −5.22*** | −6.36*** | −1.27*** | −3.58*** |
|  | (0.13) | (0.15) | (0.13) | (0.15) |
| Time*COVID lockdown | −7.34*** | −1.54*** | 24.78*** | 38.52*** |
|  | (0.13) | (0.40) | (0.28) | (0.50) |
| Second COVID lockdown |  | −5.47*** |  | −11.83*** |
|  |  | (0.36) |  | (0.35) |
| Time$^2$ |  |  | −8.48*** | −8.79*** |
|  |  |  | (0.07) | (0.07) |
| AIC | 2745927.21 | 2745700.12 | 2730539.77 | 2729429.40 |
| BIC | 2745991.30 | 2745774.89 | 2730614.54 | 2729514.86 |
| Deviance | 2745915.21 | 2745686.12 | 2730525.77 | 2729413.40 |
| Log Likelihood | −1372957.61 | −1372843.06 | −1365262.88 | −1364706.70 |
| Num. obs. | 321775 | 321775 | 321775 | 321775 |
| Num. groups: id | 42633 | 42633 | 42633 | 42633 |
| Var: id (Intercept) | 760.76 | 760.81 | 759.35 | 759.42 |
| Var: Residual | 192.35 | 192.19 | 182.14 | 181.42 |

***$p < 0.001$;

**$p < 0.01$;

*$p < 0.05$

datasets are generated. Three parameters are varied: number of time points, number of subjects and size of intervention. The score at the first time point (intercept) is drawn from a normal distribution with a mean of 170 and a variance of 25, which are the average math score and the variance at the first observation. For each time point after, a value of 16.5, which is the growth rate per year, is added to the score at the previous time point. After the generation of the growth curves noise, drawn for a normal distribution with mean zero and a variance of 25, is added to each time point. The effect size of the intervention is expressed as a percentage of the slope. The range of the parameters is further discussed below.

The chosen range for the number of time points is 6–20, with steps of 2. The number of time points in the motivating example is eight, with two tests per year for four years. To track progress students might be tested three times a year for their entire school career (seven years). This would add up to 21 time points, to ensure an even spread of time points before and after the intervention a maximum of 20 time points was chosen.

There are at least 30,000 students per time point in the motivating example, but in many cases, such quantities are not available. In case an individual school would perform analyses then there may not be more than 100 observations. To investigate such scenarios, we look at a range of 100–1,500 observations per time point with steps of 200.

In the motivating example the effect sizes for the step and slope change, relative to the general slope, varied from 10–37%. This is similar to the results of a recent meta-analysis into the effects of learning during the COVID period [52]. It estimated the pooled effect size of the learning loss to be $d = −0.17$. This can be translated into 42% of a normal school year's worth

**Table 3. Results of different ML models with an ITS design for cohort B.**

| | Model 1b | Model 2b | Model 3b | Model 4b |
|---|---|---|---|---|
| (Intercept) | 208.44*** | 208.44*** | 208.19*** | 205.64*** |
| | (0.12) | (0.12) | (0.12) | (0.13) |
| Time | 36.23*** | 36.23*** | 34.78*** | 19.92*** |
| | (0.07) | (0.07) | (0.11) | (0.22) |
| COVID lockdown | −3.80*** | −3.28*** | −5.00*** | −10.49*** |
| | (0.09) | (0.09) | (0.11) | (0.13) |
| Time*COVID lockdown | −13.47*** | −12.80*** | −9.04*** | 45.42*** |
| | (0.09) | (0.10) | (0.27) | (0.75) |
| Second COVID lockdown | | −1.67*** | | −21.95*** |
| | | (0.11) | | (0.28) |
| Time$^2$ | | | −1.16*** | −13.14*** |
| | | | (0.07) | (0.17) |
| AIC | 4014160.01 | 4013940.15 | 4013858.61 | 4007783.63 |
| BIC | 4014226.36 | 4014017.56 | 4013936.03 | 4007872.10 |
| Deviance | 4014148.01 | 4013926.15 | 4013844.61 | 4007767.63 |
| Log Likelihood | −2007074.00 | −2006963.07 | −2006922.31 | −2003883.81 |
| Num. obs. | 469455 | 469455 | 469455 | 469455 |
| Num. groups: id | 70937 | 70937 | 70937 | 70937 |
| Var: id (Intercept) | 793.20 | 793.26 | 793.21 | 794.16 |
| Var: Residual | 187.11 | 187.00 | 186.96 | 184.12 |

***$p < 0.001$;

**$p < 0.01$;

*$p < 0.05$

**Table 4. Parameters used in the simulation study.**

| Number of time points | Number of subjects | Size of intervention |
|---|---|---|
| 6 | 100 | -1% |
| 6 | 100 | -2.5% |
| 6 | 100 | -5% |
| 6 | 100 | -10% |
| 6 | 100 | -15% |
| 6 | 100 | -30% |
| 6 | 300 | -1% |
| . . .. | . . . | . . . |
| . . .. | . . . | . . . |
| 20 | 1,500 | -1% |
| 20 | 1,500 | -2.5% |
| 20 | 1,500 | -5% |
| 20 | 1,500 | -10% |
| 20 | 1,500 | -15% |
| 20 | 1,500 | -30% |

The total number of data-generating scenarios is 8*8*6 = 384.

of learning [52]. In other scenarios, the effect size might not be so large so we investigate a range of -1–30% of the pre-intervention slope; Full range: -1%, -2.5%, -5%, -10%, -15%, -30%. In line with the motivating example, both the step change and the slope change are defined as negative shocks. Additionally, the effect size is set equal for the step and slope change in case both are specified in the data.

Furthermore, interventions can have different kinds of impacts. Simulation scenarios with only a step or slope change as well as those with both are considered. It could even be the case that there is an initial step change, followed by a smaller step change later, as was the case in the motivating example. This simulation scenario is indicated by the two steps and slope. These four simulation scenarios, visualised in Fig 2, are investigated. For each simulation scenario, 384 datasets are simulated.

The timing of the intervention will be in the middle of the time series, as an early or late intervention has little effect on the power in an ITS design [24]. This means that the event will take place between two time points, as we can see in Fig 2. This allows for an even spread of time points before and after the intervention. In the case of a secondary intervention, the timing is set at 75% of the time series. This does not leave more than two time points in between and after the two interventions in all scenarios. Because of this, the slope change was held constant after the first intervention The value of the secondary step change is set to 50% of the original step change.

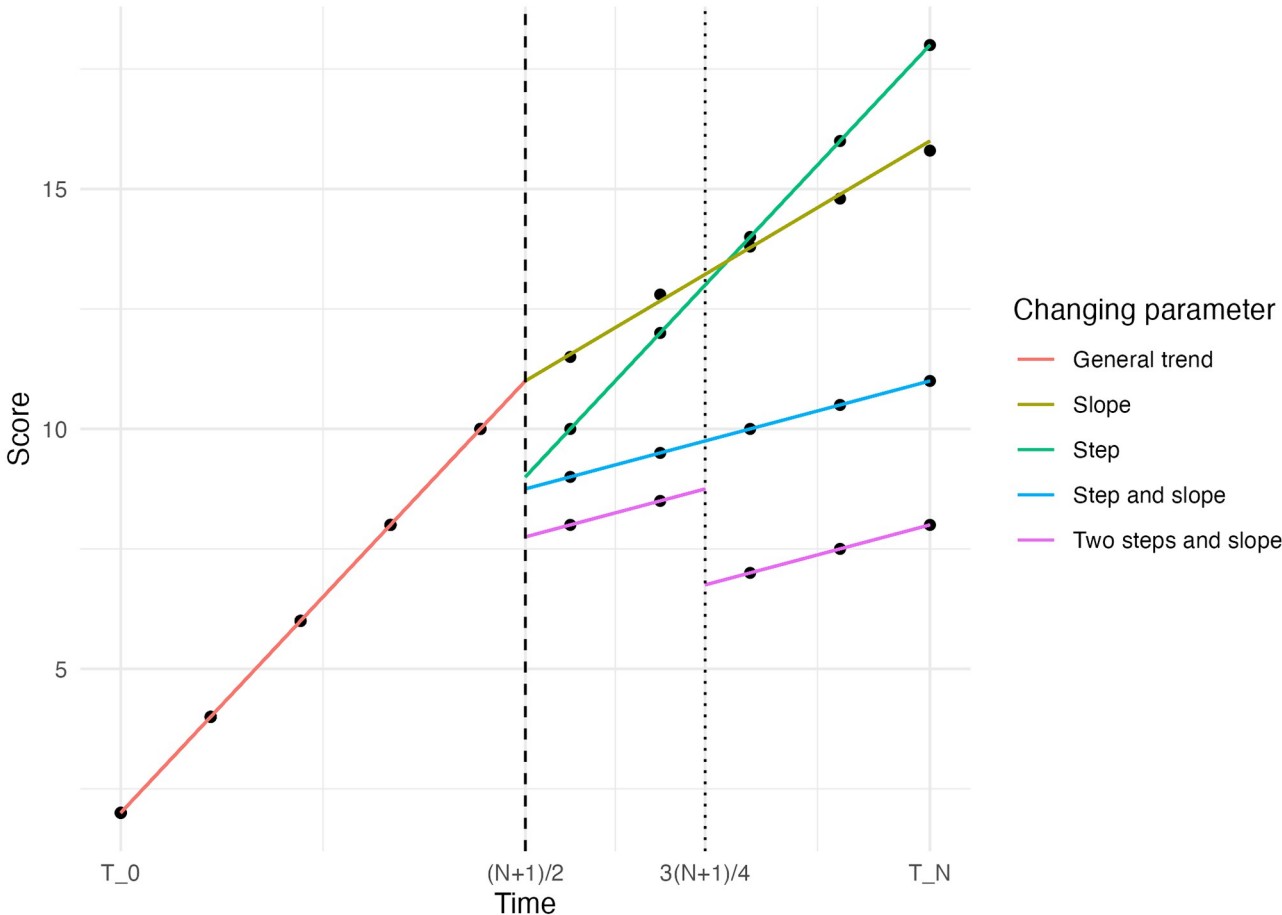

**Fig 2. The simulation situations.** The dashed line indicates the first intervention and the dotted line the second intervention.

## Methods of analysis

Table 5 shows the simulation scenarios and the models used to estimate the ITS design.

In the first simulation scenario, the model matches the data generated. In the next two models, the model is an overfit to the data, due to there not being a slope change, and there not being a step change. The data in the following simulation scenario adds a second intervention, while this is not specified in the model so there is an underfit. In these three situations the basic ITS design Eq 2 is used. This is followed by a simulation scenario where the second intervention is correctly specified. Lastly, there is a simulation scenario with a parameter for time squared in the model, but not in the data (overfitting).

These models resemble models that researchers might want to investigate, as was the case in the motivating example. The effect of under and overfitting is considered since in practise it is impossible to know if the model will have the right fit to the data.

## Performance

The performance was based on three criteria: power, bias and precision. The empirical power to reject the null hypothesis will be calculated as the proportion of simulations, where the p-value for a coefficient was $< 0.05$. The bias is the absolute difference between the observed estimated and the defined parameter. Precision is defined as the standard error of the estimate.

All analysis are performed using R 1.4.1717 [53]. The code for the simulation study can be found on https://github.com/Fdvanleeuwen/ITS_designs_Simulation.

## Results

The performance of the ITS design is discussed per simulation scenario in the order of Table 5. The most relevant figures for the power, bias and precision are shown in the main text. The performance measures are discussed in the performance section. The heatmaps indicate the performance of the models, as can be inferred by the colour, for different scenarios of the number of time points, sample size and effect size. Using heatmaps allows for seeing patterns in the results.

## Step and slope change

In this scenario, there was a step and slope change in the data. In Fig 3 it is evident that the power is high if the effect size (compared to the pre-level slope) is above 0.30. Adequate power ($> .80$) was observed when the effect size was 0.15, the sample size was 1,100 and the number of time points was 12. With an effect size of 0.3, the sample size was 500 and the number of time points was eight for adequate power.

**Table 5. Different models and simulation scenarios.**

| Model | Data | Simulation Scenario |
|---|---|---|
| $\hat{Y}_t = \beta_0 + \beta_1 T + \beta_2 X_t + \beta_3 TX_t$ | Step and slope | Right fit |
| $\hat{Y}_t = \beta_0 + \beta_1 T + \beta_2 X_t + \beta_3 TX_t$ | Only step | Overfit |
| $\hat{Y}_t = \beta_0 + \beta_1 T + \beta_2 X_t + \beta_3 TX_t$ | Only slope | Overfit |
| $\hat{Y}_t = \beta_0 + \beta_1 T + \beta_2 X_t + \beta_3 TX_t$ | Two steps and slope | Underfit |
| $\hat{Y}_t = \beta_0 + \beta_1 T + \beta_2 X_{1t} + \beta_3 X_{2t} + \beta_4 TX_t$ | Two steps and slope | Right fit |
| $\hat{Y}_t = \beta_0 + \beta_1 T + \beta_2 X_t + \beta_3 TX_t + \beta_4 T^2$ | Step and slope | Overfit |

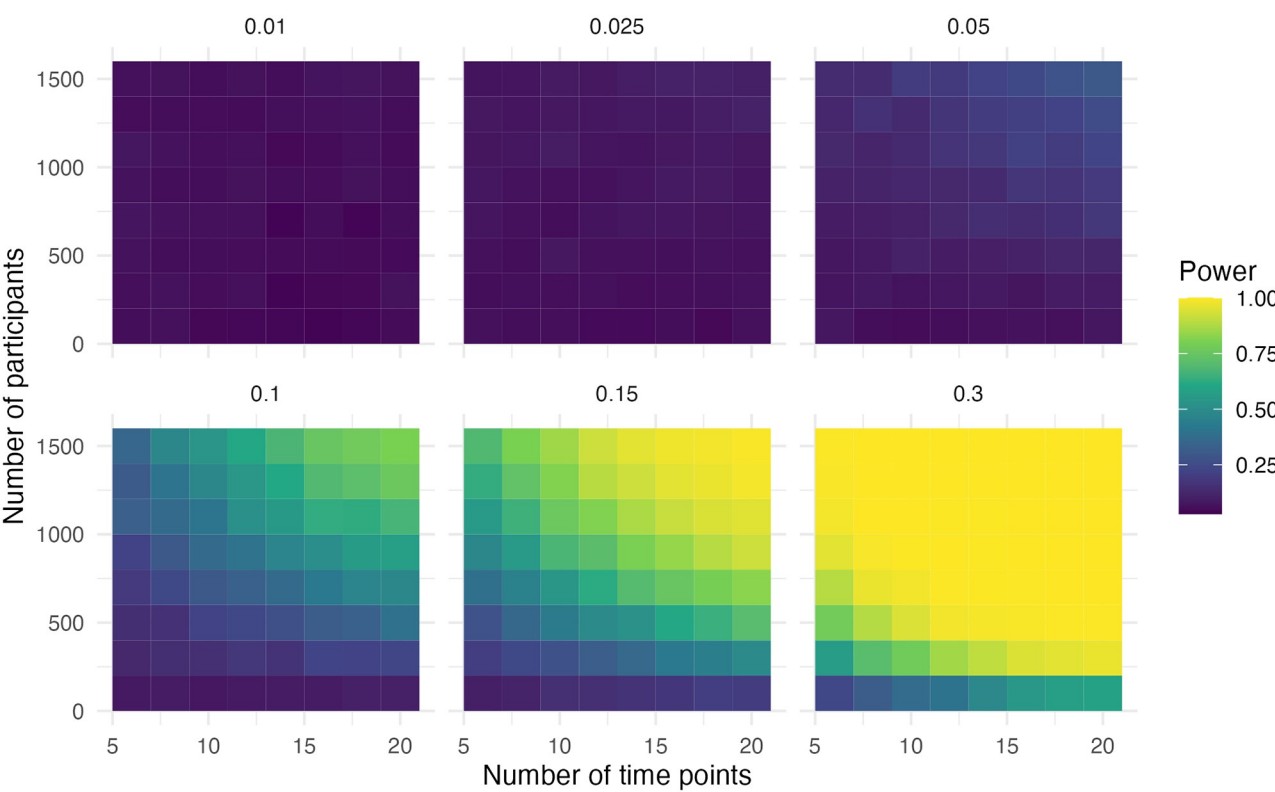

**Fig 3. Power of the step for simulation scenario with a step and slope change.**

In Fig 4 we can observe that adequate power (> .80) was observed when the effect size was 0.025, the sample size was 1,100 and the number of time points was 18. With an effect size of 0.05, the sample size was 1,300 and the number of time points was 10 for adequate power. With an effect size of 0.1, the sample size was 700 and the number of time points was eight for adequate power. With an effect size of 0.15, the sample size was 700 and the number of time points was six for adequate power. With an effect size of 0.3, the sample size was 300 and the number of time points was eight for adequate power. Scenarios, where the step change and slope change had opposing signs, resulted in the same power.

## Only slope or step change

In both these scenarios there is only one parameter specified in the data, but the full segmented regression equation is used. The power for the non-specified parameter is close to 0.05 as expected (S1 and S2 Figs). The power for the parameter specified in the data is the same as in the scenario with two parameters specified in the data (S3 and S4 Figs).

## Step and slope change and additional step change

In this simulation scenario, the second step is defined as 50% of the first step change. The power is similar to the first scenario with a step and slope change where there was only one intervention specified (S5 and S6 Figs). If there is no correction for such a second intervention, then the results can become biased. The step change is overestimated (Fig 5). There is a pattern in the bias of the step that is caused by the location of the second intervention. The number of

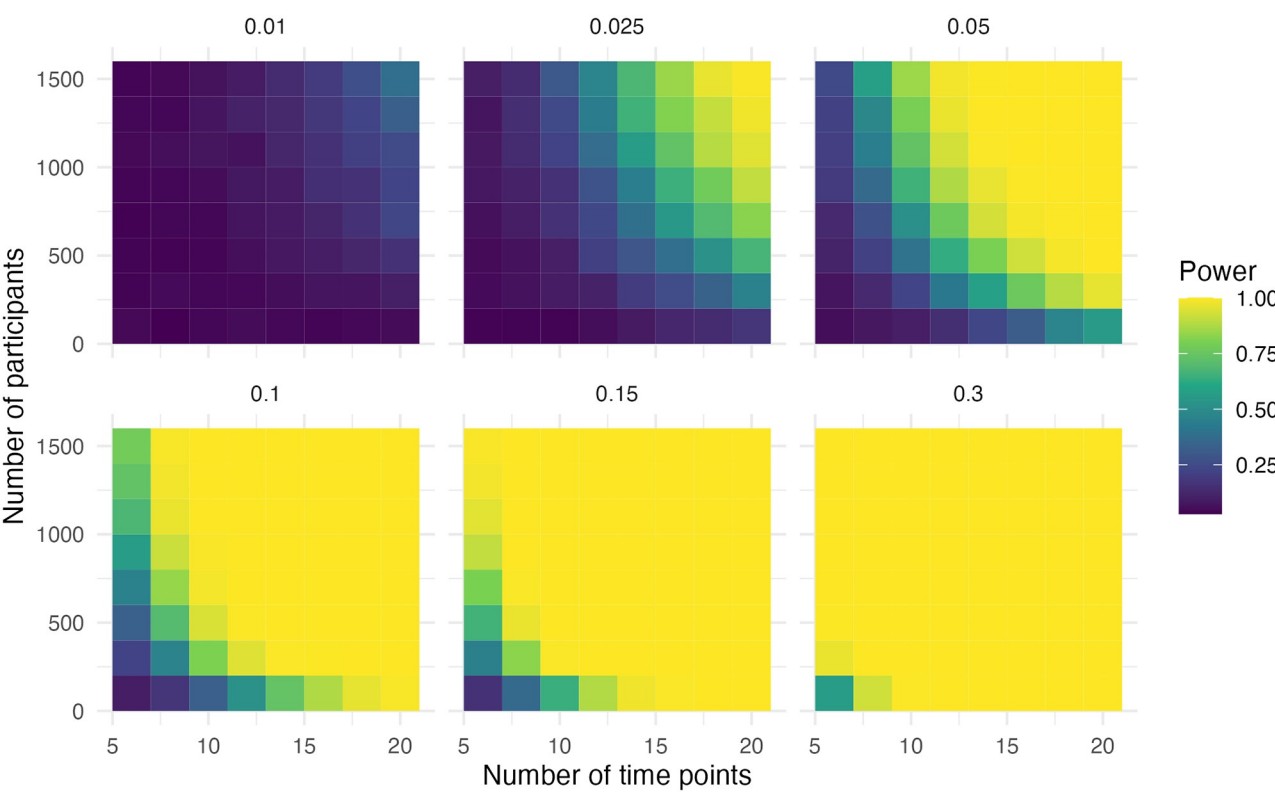

**Fig 4. Power of the slope for simulation scenario with a step and slope change.**

time points between the interventions and after the second intervention alternates. In the scenario with six time points, there is no observation between the interventions and one observation after the second intervention (0/1). In the scenario with eight time points, there is an observation between the interventions (1/1). This sequence continues in 1/2, 2/2, etc. It seems that the bias in the step is larger when there are more observations between the interventions, but this effect can be compensated for by having more observations after the second intervention. This can be seen in relatively lower bias at six time points, the increase at eight time points and then a decrease again at ten time points. The bias in the step decreases overall as the number of time points increases.

The slope change is underestimated (Fig 6). The amount of underestimation in the slope change depends on the number of time points, the more time points the less biased the estimate is as the model has the possibility to detect catch-up effects. The absolute amount of bias increases as the effect size goes up.

### Step and slope change and additional step change with correction

The data in this scenario is the same as in the previous scenario, but now there is a correction for the extra intervention. In Fig 7 it can be seen that adequate power (> .80) was observed when the effect size was 0.15, the sample size was 1,500 and the number of time points was ten. With an effect size of 0.3, the sample size was 700 and the number of time points was six for adequate power. Both are very similar to the first scenario with a step and slope change.

In Fig 8, we can observe that adequate power (> .80) was observed when the effect size was 0.05, the sample size was 1,300 and the number of time points was 14. In the case of an effect

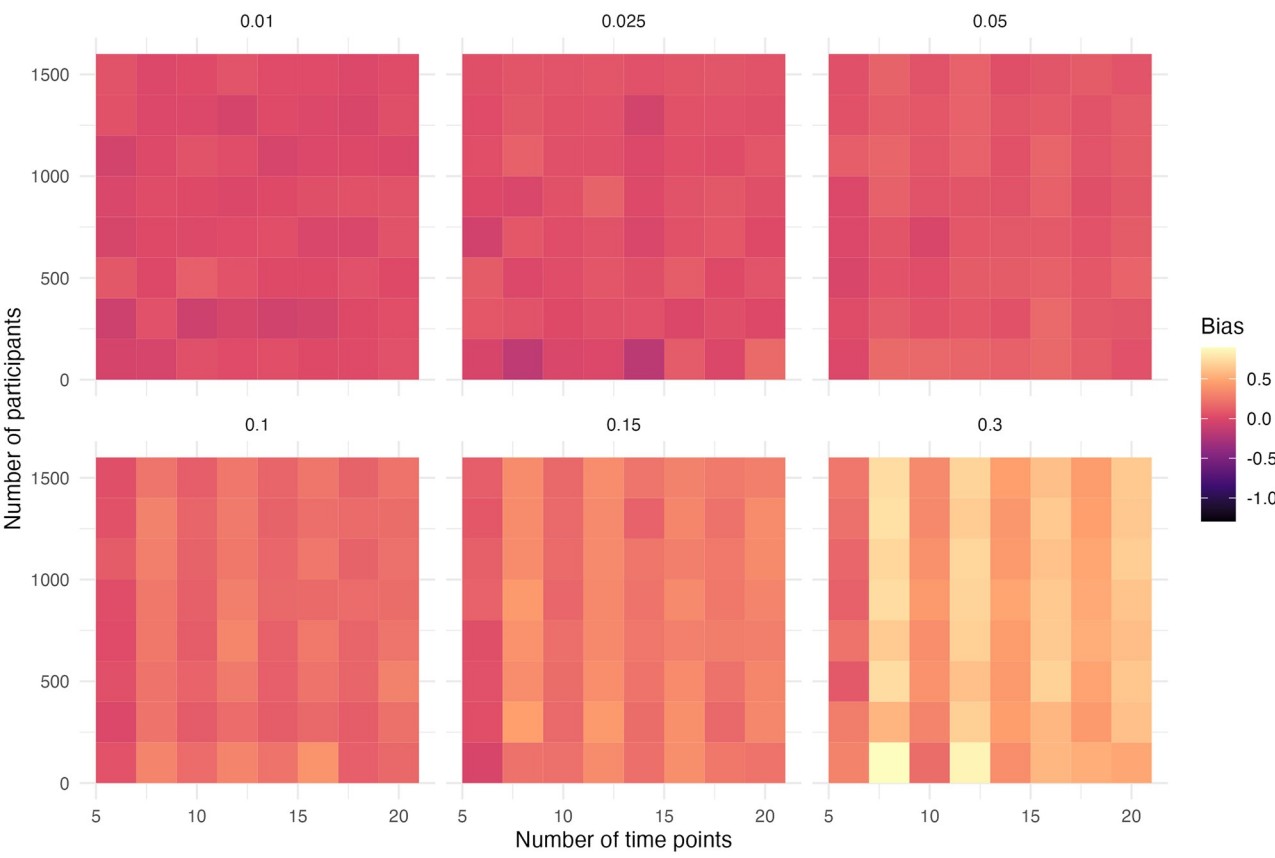

**Fig 5. Absolute bias of the step for simulation scenario with a step and slope change and an additional step change.**

size of 0.1, the sample size was 900 and the number of time points was ten for adequate power. With an effect size of 0.15, the sample size was 900 and the number of time points was eight for adequate power. With an effect size of 0.3, the sample size was 500 and the number of time points was six for adequate power. This is less than the first scenario with a step and slope change.

The power for the second step change is relatively low, which is expected as the effect size is 50% of the step. Adequate power was only observed with an effect size of 0.15, a sample size of 900 and the number of time points was six (Fig 9). There is again a pattern due to the rounding of the timing of the second intervention.

## Step and slope change with correction for a changing slope

In this scenario, a squared term is introduced to the model to investigate a case where there might be a changing slope. The decision to include this parameter can be made based on the literature, as was the case in the motivating example. There is no change in the slope over time specified in the data-generating process, except the slope change. The power for the step change is similar to that of the ITS design without the time-squared term in the first scenario with a step and slope change (Fig 10). The power of the slope is lower (Fig 11). Adequate power was observed when the effect size was 0.15, the sample size was 1,400 and the number of time points was 14. With an effect size of 0.3, the sample size was 700 and the number of time

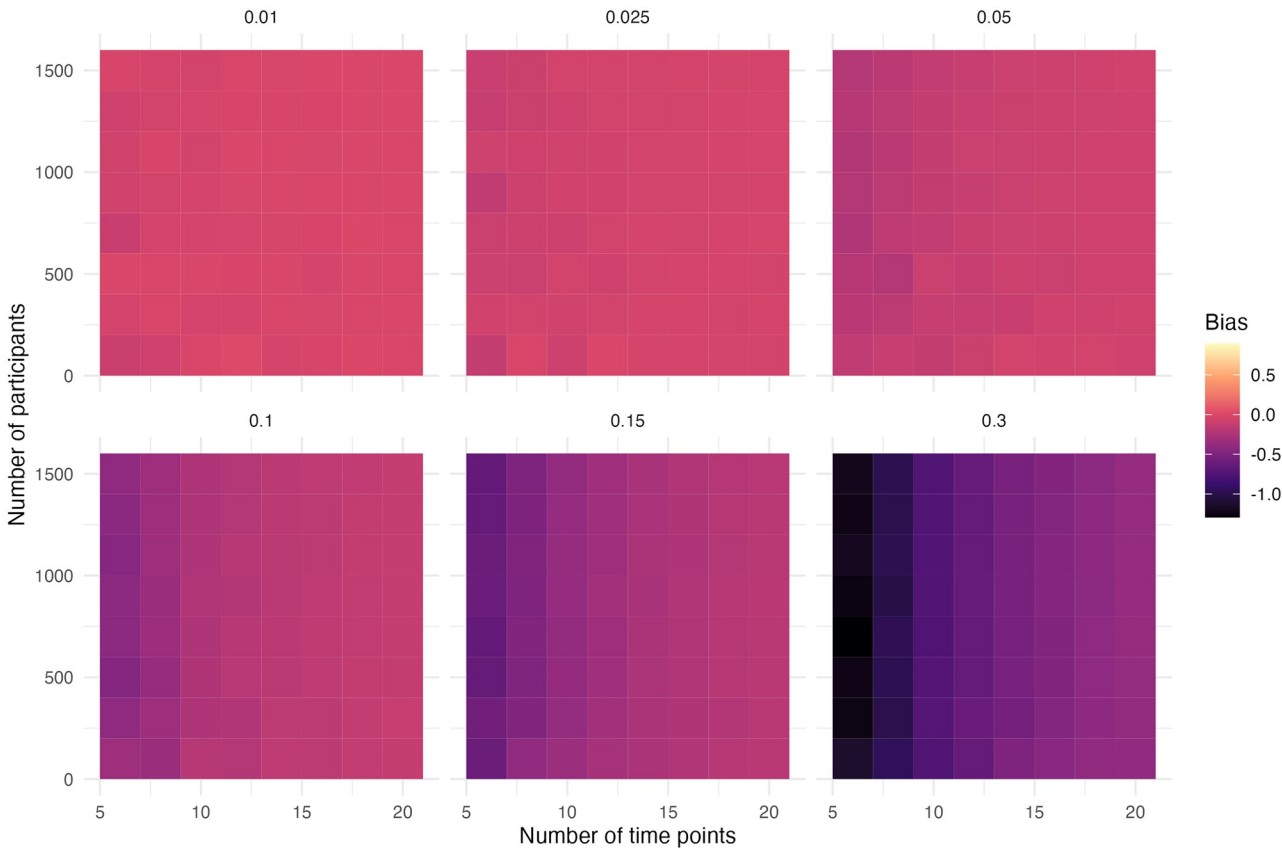

**Fig 6. Absolute bias of the slope for simulation scenario with a step and slope change and an additional step change.**

points was ten for adequate power. Additionally, the bias, as well as the precision in the slope, was much higher (S7 and S8 Figs).

## Discussion

This research has investigated the performance of the standard ITS design as well as that of various extensions, in situations with limited data. While the ITS design works well under many different circumstances, there are some factors to consider before running an analysis: the sample size in the case of a step change, the number of time points for a slope change, and the problem of underfitting\overfitting. Not one of these factors alone can account for the power, it always depends on the a combination of factors.

In the basic scenario with both a step and a slope change the expected power is as follows: when a step change of 0.15 (% of the slope) is expected, the required sample size is about 1,100 with a minimum of twelve time points. In the case of an effect size of 0.3, the required sample size decreases to 500 with eight time points. For detecting a slope change of 0.1 the minimum number of time points necessary is 20 and a minimum sample size of 100. If the expected effect size is larger, 0.15 or 0.3, the number of time points needed decreases to 14 and 10 respectively. Detecting a change in slope can be difficult if there are few time points and if the effect size is small. This is because there is sampling error at every time point and increasing the number of time points smooths these errors out.

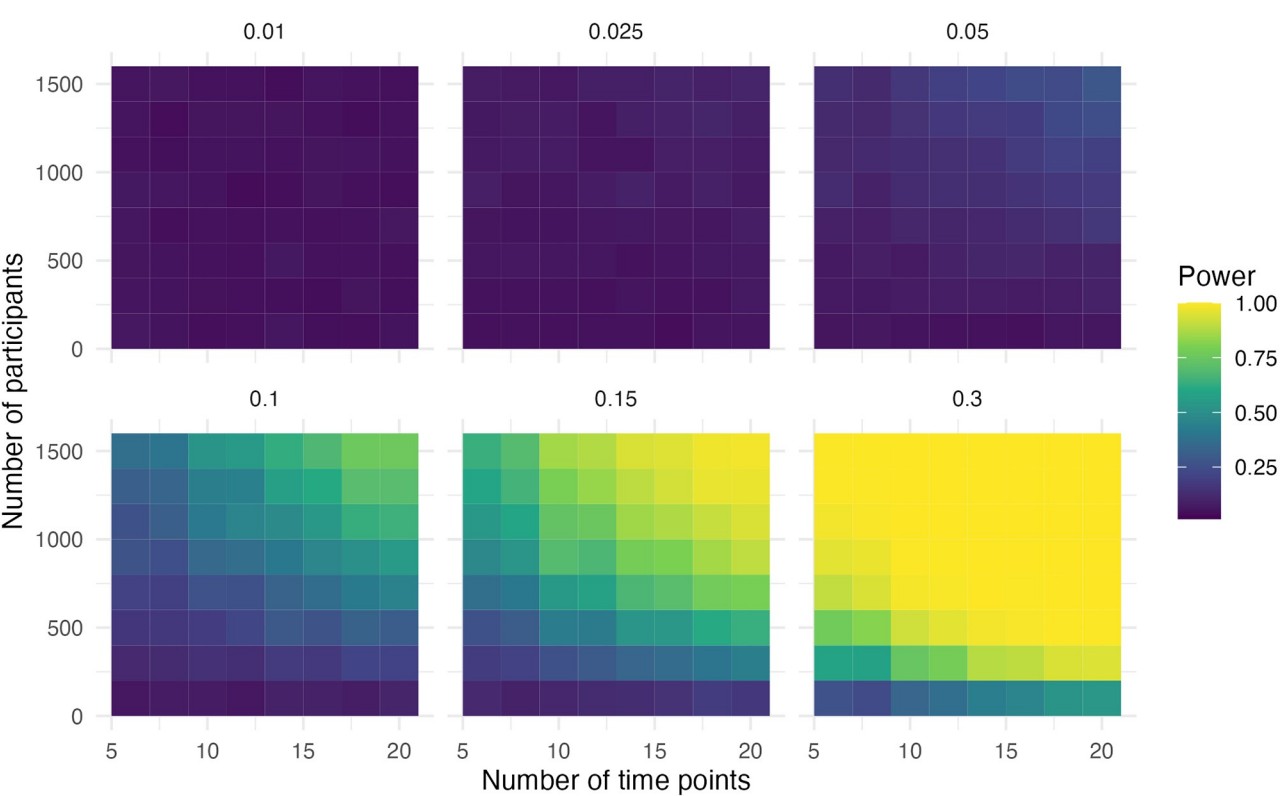

**Fig 7. Power of the step for simulation scenario with a step change, a slope change and an additional step change with correction.**

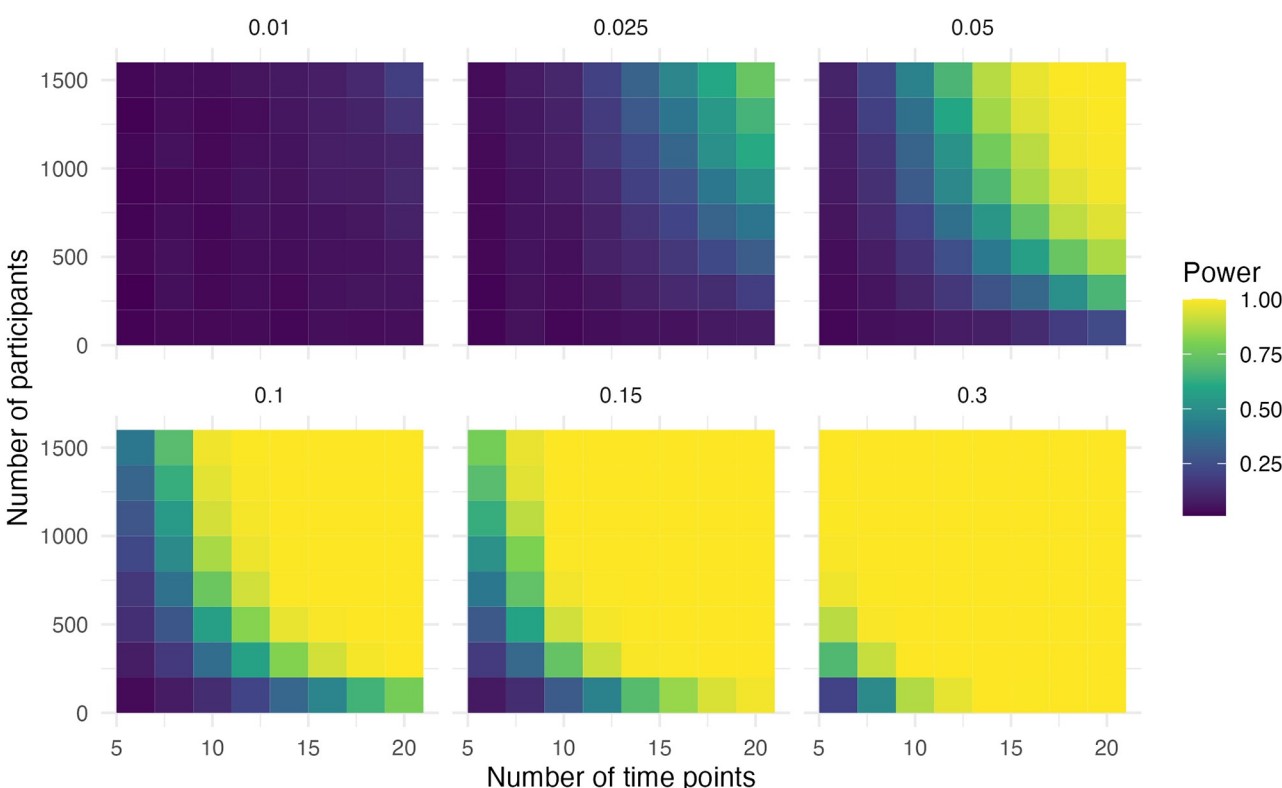

**Fig 8. Power of the slope for simulation scenario with a step change, a slope change and an additional step change with correction.**

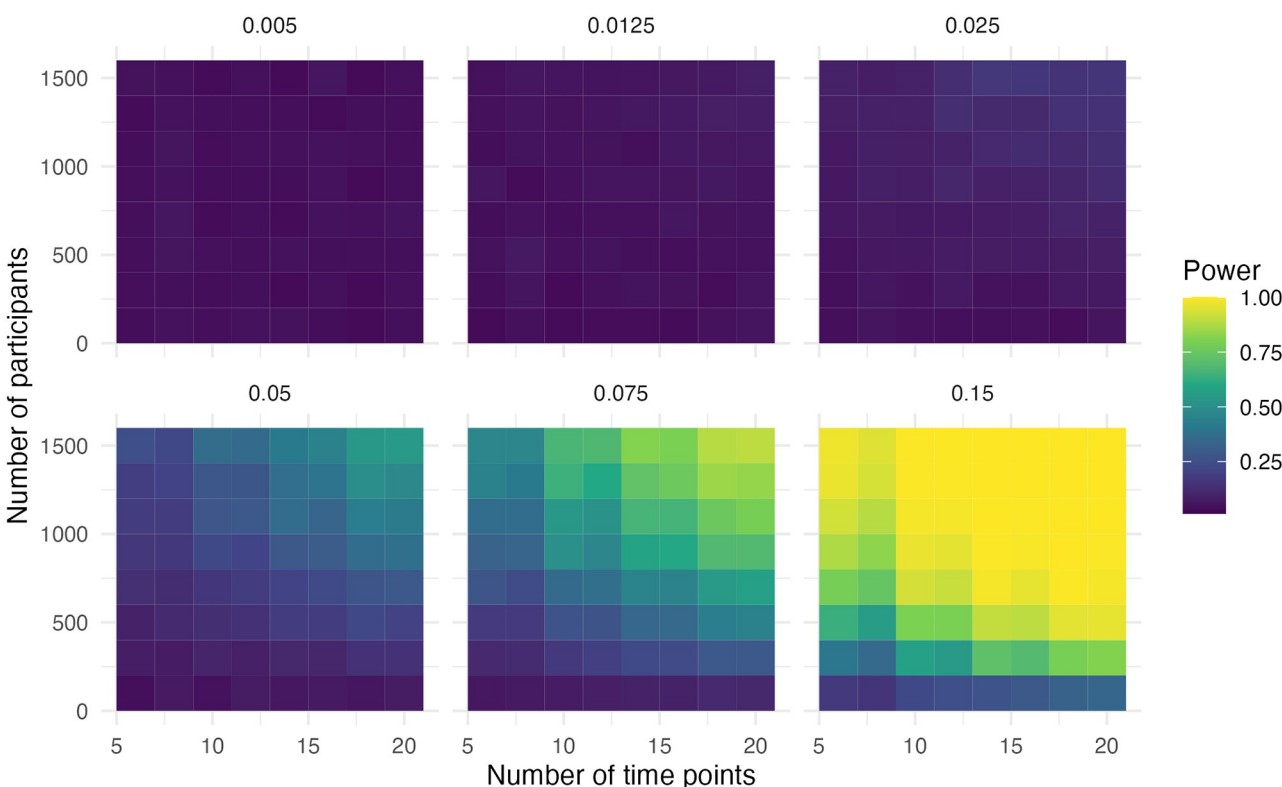

**Fig 9. Power of the additional step for simulation scenario with a step change, a slope change and an additional step change with correction.**

Overfitting can severely decrease the performance of the ITS design. By including a term for time squared, the power of the change in slope parameter declines. It is thus important to hypothesize which factors play a role in the time series, and, if this role is large enough, to include an extra parameter in the model. In practice, overfitting is not a big problem as most analyses are performed with a standard ITS design including only a step and slope change.

Underfitting is also a larger obstacle for ITS designs. Using the standard ITS design in scenarios which are very complicated, e.g. the motivating example, can result in biased estimates. As demonstrated in the scenario with a secondary intervention and no correction, both the step change and the slope change were biased. This can be solved by adding another parameter to divide the time points into one more part per extra intervention. Such an extra parameter does decrease the power slightly for the first step change and for the slope change. It is important to note that there need to be at least two time points in every part to be able to detect a change in the slope.

Our findings show that power is attainable with a small number of time points for both a step and slope change. Research concerning ITS designs, with data that is similar to the motivating example, can expect higher power than was observed in the simulation study of Hawley et al. [24] concerning incidence rates. The lower power found is likely caused by more variance in two parameters: the outcome variable and the sample size over time. In this study, the latter was held constant over all time points, while Hawley et al. [24] drew the sample size from a normal distribution with the assigned sample size (n) as the mean and an SD of n/3.

The present focus on a low number of time points has demonstrated that it is not always necessary to have 6, 8 or 12 time points before and after the intervention, as is prescribed

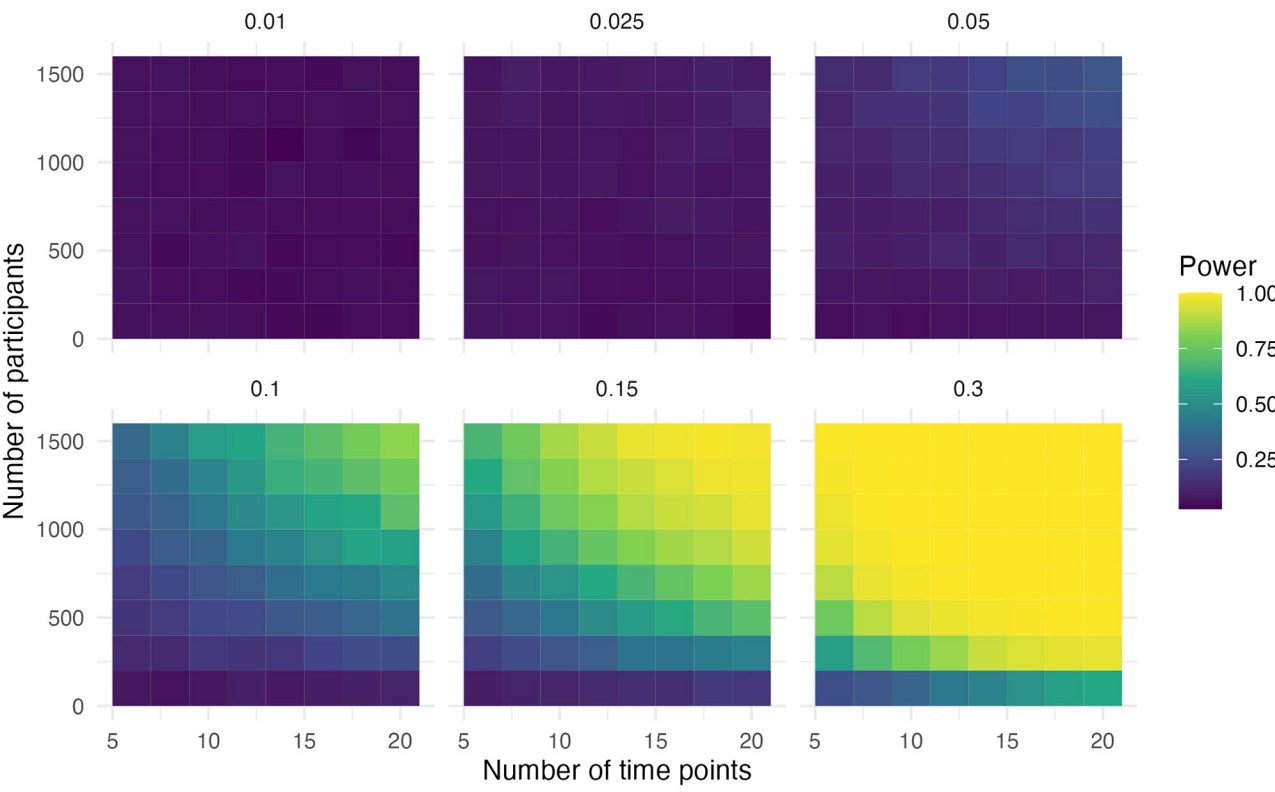

**Fig 10. Power of the step for simulation scenario with a step change, a slope change and a correction for changing slope.**

in the literature [17–19]. Moreover, extended ITS designs performed well with a low number of time points and a small sample size as long as the effect size was relatively large. The findings of this research enable more researchers to consider an ITS analysis with small data sets.

In the motivating example, the location of the intervention seemed to have a great impact on the stability of the estimates. Cohort A (17/18) only had three time points after the first intervention to model two-step changes and one slope change. Cohort B (18/19) had five time points after the first intervention, and this helped with the stability. Choosing the cohort has implications for the estimates. Researchers should consider the location of the intervention(s) in the time series when deciding which model to use. It is almost never the case that the intervention is exactly in the middle of a time series. If the data come from regular surveys then there is probably more data before the intervention than after, as research is often interested in changes that happened not long ago. Another case could be that the observations only start close to the intervention. For example, a policy shift is planned for next year, and to monitor the projected effect data collection starts. In both cases, it is important to have sufficient data to estimate a general trend before the intervention, else the estimated effect of the intervention relies on a weak counterfactual.

In the motivating example, both cohorts had sufficient data to estimate the general trend, but only cohort B had sufficient data after the intervention to model all effects properly. The estimates from cohort B are used for inferences. In the stable models, the effect of the first step change ranged between -9% and -14% of the slope. This can be translated into a learning loss of four to six weeks.

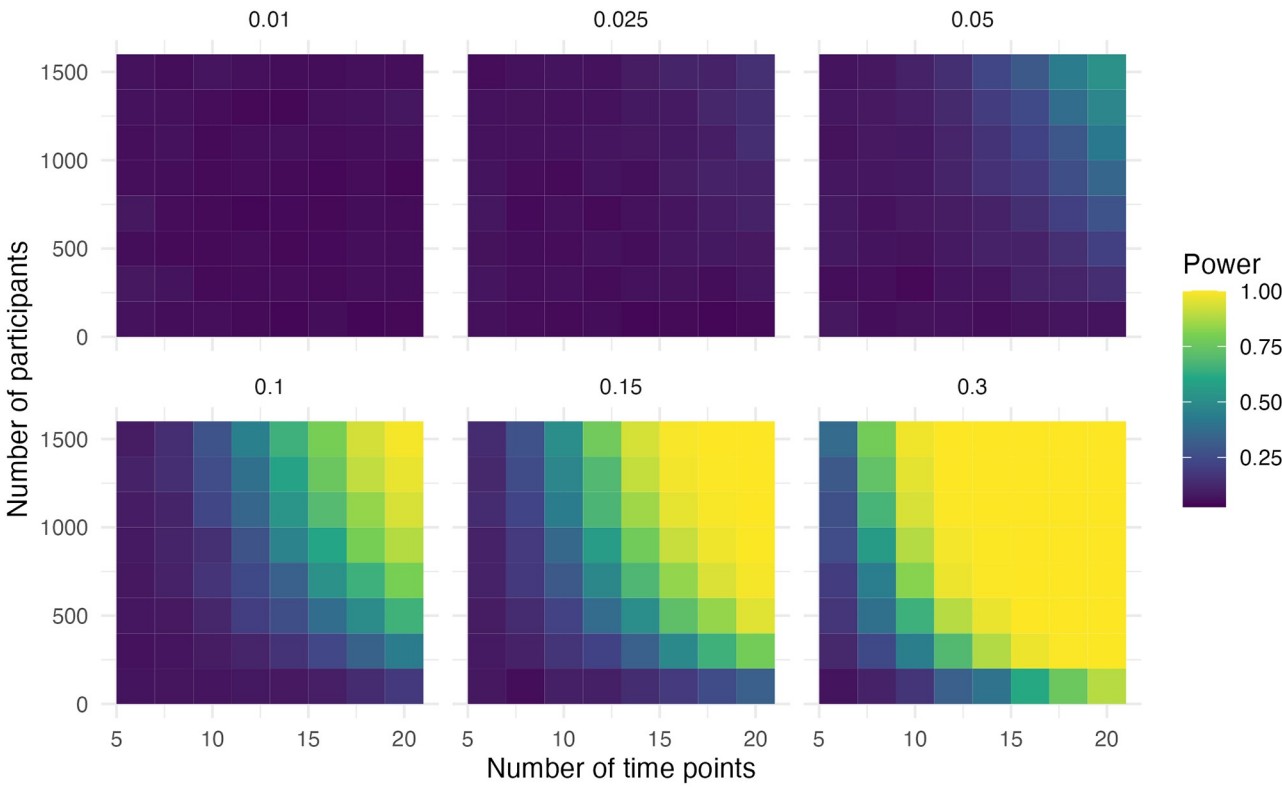

**Fig 11. Power of the slope for simulation scenario with a step change, a slope change and a correction for a changing slope.**

For the slope change the scenario with a correction for a decrease in learning rate (time squared) seems to be the most appropriate model, based on the literature. The slope change was -25% and would suggest the learning rate of students decreased by a quarter after the first lockdown. The model with the second intervention specified indicated that there was a small positive effect on math scores of 4%. The results of the step change are similar to those of Engzell et al. [4] in their study of learning losses in the Netherlands. They found the effect of the first lockdown to be equal to eight weeks of learning. They used a difference-in-difference design, making it impossible to compare the effect of the slope change.

One limitation of this study is that the results cannot be interpreted as causal effects, as the basic ITS design is not able to exclude confounding events occurring at a similar time as the interventions. A control group, if available, might sometimes be introduced to combat this threat to external validity, although this is not possible in many situations. Another limitation is that only one value of the variance in the outcome variable was examined and, moreover, it was held constant over time. This represented the motivating example, but might not be applicable in other scenarios.

Future work can focus on the performance of the extended ITS designs with multilevel models, piecewise latent growth models or autocorrelation models. Scenarios, where the effect size has a different effect size for the step change and slope change, can also be explored. Additionally, the ITS design can be embedded in causal modelling using directed acyclic graphs (DAGs) or priors can be included using a Bayesian framework. This should strengthen the causal claims of ITS designs. Moreover, the performance of even more complex, non-linear, models could be tested.

## Conclusion

In a nutshell, we conclude that the interrupted time series design is a strong method for assessing changes over time, even with few time points. This paper shows that the notion that many time points are always necessary for an ITS analysis is incorrect. Additionally, this paper shows that employing ITS designs can be difficult when the data are complex, such as multiple interventions or non-linear time effects.

These findings are mostly important for researchers who design longitudinal studies, and have to think about how to space waves of data collection. Also, people who want to analyse existing longitudinal data with an interrupted time series design can use the findings from our simulation study to understand whether their study, given an expected or observed effect size has enough statistical power to estimate a causal effect reliably.

## Supporting information

**S1 Fig. Power of the step for simulation scenario with only a slope change.**
(TIF)

**S2 Fig. Power of the slope for simulation scenario with only a step change.**
(TIF)

**S3 Fig. Power of the step for simulation scenario with only a step change.**
(TIF)

**S4 Fig. Power of the slope for simulation scenario with only a slope change.**
(TIF)

**S5 Fig. Power of the step for simulation scenario with a step and slope change and a additional step change.**
(TIF)

**S6 Fig. Power of the slope for simulation scenario with a step and slope change and a additional step change.**
(TIF)

**S7 Fig. Absolute bias of the slope for simulation scenario with a step and a slope change and a correcting for a changing slope.**
(TIF)

**S8 Fig. Precision in SE of the slope for simulation scenario with a step and a slope change and a correcting for a changing slope.**
(TIF)

## Acknowledgments

The authors would like to thank to Marco van Leeuwen and Alex Carriero for helpful comments. Earlier version of this paper was presented at the 2022 IRT workshop at Twente University.

## Author Contributions

**Conceptualization:** Florian D. van Leeuwen, Peter Lugtig, Remco Feskens.

**Data curation:** Remco Feskens.

**Formal analysis:** Florian D. van Leeuwen.

**Investigation:** Florian D. van Leeuwen.

**Methodology:** Florian D. van Leeuwen.

**Supervision:** Peter Lugtig, Remco Feskens.

**Visualization:** Florian D. van Leeuwen.

**Writing – original draft:** Florian D. van Leeuwen.

**Writing – review & editing:** Florian D. van Leeuwen, Peter Lugtig, Remco Feskens.

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
