## [Decision Letter · Decision Letter 0]

13 Feb 2024

PONE-D-23-39424The performance of interrupted time series designs with a limited number of time points: Learning losses due to school closures during the COVID pandemicPLOS ONE

Dear Dr. van Leeuwen,

Thank you for submitting your manuscript to PLOS ONE. After careful consideration, we feel that it has merit but does not fully meet PLOS ONE’s publication criteria as it currently stands. Therefore, we invite you to submit a revised version of the manuscript that addresses the points raised during the review process.

We look forward to receiving your revised manuscript.

Kind regards,

Sathishkumar Veerappampalayam Easwaramoorthy

Academic Editor

PLOS ONE

Journal Requirements:

5. Please ensure that you refer to Figure 11 in your text as, if accepted, production will need this reference to link the reader to the figure.

Reviewers' comments:

Reviewer's Responses to Questions

**Comments to the Author**

1. Is the manuscript technically sound, and do the data support the conclusions?

Reviewer #1: Yes

Reviewer #2: Yes

2. Has the statistical analysis been performed appropriately and rigorously? 

Reviewer #1: Yes

Reviewer #2: Yes

3. Have the authors made all data underlying the findings in their manuscript fully available?

Reviewer #1: Yes

Reviewer #2: Yes

4. Is the manuscript presented in an intelligible fashion and written in standard English?

Reviewer #1: Yes

Reviewer #2: Yes

5. Review Comments to the Author

Reviewer #1: This is a nice study examining the potential of ITS on naturalistic settings with the covid-19 pandemic as an example. I would recommend emphasizing the key contribution of this study in more detail in the abstract. It is nice that the authors highlight the potential of the ITS for such settings as other analyses such as the difference-in-difference approach also have their limitations.

The article is overall well-written and I did not detect potential flaws. AS such, I have no more concrete points to add.

Reviewer #2: The interrupted time series (ITS) have been addressed in this study. The Monte Carlo simulation method has been selected to serve the proposed issues. Some results collected have been presented in this study. This study is interesting however there are some drawbacks that the authors should address them to improve this study.

1.There are a lot of typing mistakes which have been found in this paper.

2.The main contribution of this study must be provided.

3.The proposed models must be indicated clearly.

4.The evaluation and analysis of collected results must be indicated clearly.

5.The significance of this study must be provided in the conclusion part.

6. PLOS authors have the option to publish the peer review history of their article (what does this mean?). If published, this will include your full peer review and any attached files.

Reviewer #1: No

Reviewer #2: No

---

## [Author Response · Author response to Decision Letter 0]

11 Mar 2024

Point-by-point reply letter to reviewer comments of article PONE-D-23-39424

“The performance of interrupted time series designs with a limited number of time points: Learning losses due to school closures during the COVID-19 pandemic”

We want to thank the (associate) editor(s) and anonymous reviewers for their comments on our earlier version of the manuscript “The performance of interrupted time series designs with a limited number of time points: Learning losses due to school closures during the COVID-19 pandemic”.

In this letter we provide a point-by-point reply to the comments provided, and indicate how we have changed the main text of the paper. The comments have been helpful, especially in putting more emphasis on the main contribution of the paper, and several stylistic and textual changes. We are confident the manuscript has been improved, and are looking forward to see this manuscript published in PLos One.

Comments by the (associate) editor(s):

2. We suggest you thoroughly copyedit your manuscript for language usage, spelling, and grammar. If you do not know anyone who can help you do this, you may wish to consider employing a professional scientific editing 

All authors went through the manuscript again, and have made about 150 changes to the manuscript in terms of spelling, style and grammar. We attach a version of the manuscript with tracked changes, so it is easy to see what we changed. The manuscript has benefited from this. 

We obtained written consent from the ethical review board. We have added this to the text, and along with the registration number for this study, it is possible to locate our ethical review at Utrecht University.

We have added this as well.

We have added the ORCID ID “0009-0009-7092-2848” for the corrspondinng author into Editrial Manager. 

5. Please ensure that you refer to Figure 11 in your text as, if accepted, production will need this reference to link the reader to the figure.

We have added this reference to Figure 11 to the text.

Comments by Reviewer 1:

This is a nice study examining the potential of ITS on naturalistic settings with the covid-19 pandemic as an example. I would recommend emphasizing the key contribution of this study in more detail in the abstract. It is nice that the authors highlight the potential of the ITS for such settings as other analyses such as the difference-in-difference approach also have their limitations.

The article is overall well-written and I did not detect potential flaws. AS such, I have no more concrete points to add.

We thank the reviewer for these comments. We have rewritten parts of the abstract, introduction, background and discussion sections to put more emphasis on the contribution of this article, and the potential of the ITS model, following similar comments by reviewer#2.

 

Comments by Reviewer 2:

 The interrupted time series (ITS) have been addressed in this study. The Monte Carlo simulation method has been selected to serve the proposed issues. Some results collected have been presented in this study. This study is interesting however there are some drawbacks that the authors should address them to improve this study.

1.There are a lot of typing mistakes which have been found in this paper.

All authors went through the manuscript again, and have made about 50 changes to the manuscript in terms of spelling, style and grammar. We attach a version of the manuscript with tracked changes, so it is easy to see what we changed. The manuscript has benefited from this. 

2.The main contribution of this study must be provided.

Here, we made several changes to the abstract (based on #R1 comments), the introduction, background and discussion to highlight the main contribution (under what circumstances can the ITS be used). The version with tracked changes provides can be used to easily see these changes.

3.The proposed models must be indicated clearly.

Equations 1 to 5 show the basic ITS model, along with several variations, all building on the standard regression model in equation 1, so that it is easy to readers familiar with regression what the ITS model does. In table 5 we have indicated the specific simulation scenario’s and model used to evaluate the ITS. These models in terms of notation directly relate to equations 1 to 5. We hope this clarifies for the reviewer the models we use. 

4.The evaluation and analysis of collected results must be indicated clearly.

We did not alter the results section greatly (apart from grammar and spelling), but we did add a “reading guide” at the start of the results to guide readers through this section:

“The performance of the ITS design is discussed per simulation scenario in the order of Table 5. 

The most relevant figures for the power, bias and precision are shown in the main text. The performance measures are discussed in the performance section. The remainder of the figures can be found in the appendix. The heatmaps indicate the performance of the models, as can be inferred by the colour, for different scenarios of the number of time points, sample size and effect size. Using heatmaps allows for seeing patterns in the results.”

5.The significance of this study must be provided in the conclusion part.

This comment is related to comment 2. We have changed the last part of this section to highlight the significance of the study:

“In a nutshell, we conclude that the interrupted time series design is a strong method for assessing changes over time, even with few time points. This paper shows that the notion that many time points are always necessary for an ITS analysis is shown to be incorrect. Additionally, this paper shows that employing ITS designs can be difficult when the data are complex, such as multiple interventions or non-linear time effects. 

The results from the simulation study can be used as a guide for future research. Researchers should thus inspect their data, hypothesize about effect sizes, and consider multiple models to be used for estimation to decide if there is enough power.”

---

## [Decision Letter · Decision Letter 1]

29 Apr 2024

PONE-D-23-39424R1The performance of interrupted time series designs with a limited number of time points: Learning losses due to school closures during the COVID-19 pandemicPLOS ONE

Dear Dr. van Leeuwen,

Thank you for submitting your manuscript to PLOS ONE. After careful consideration, we feel that it has merit but does not fully meet PLOS ONE’s publication criteria as it currently stands. Therefore, we invite you to submit a revised version of the manuscript that addresses the points raised during the review process. **Please address the remaining comments from the reviewers. Specifically, justifications of why the chosen method is superior to other alternative approaches should be carefully discussed/quantified.**

We look forward to receiving your revised manuscript.

Kind regards,

Chenfeng Xiong

Academic Editor

PLOS ONE

Reviewers' comments:

Reviewer's Responses to Questions

**Comments to the Author**

1. If the authors have adequately addressed your comments raised in a previous round of review and you feel that this manuscript is now acceptable for publication, you may indicate that here to bypass the “Comments to the Author” section, enter your conflict of interest statement in the “Confidential to Editor” section, and submit your "Accept" recommendation.

Reviewer #1: All comments have been addressed

Reviewer #2: (No Response)

2. Is the manuscript technically sound, and do the data support the conclusions?

Reviewer #1: Yes

Reviewer #2: (No Response)

3. Has the statistical analysis been performed appropriately and rigorously? 

Reviewer #1: Yes

Reviewer #2: (No Response)

4. Have the authors made all data underlying the findings in their manuscript fully available?

Reviewer #1: Yes

Reviewer #2: (No Response)

5. Is the manuscript presented in an intelligible fashion and written in standard English?

Reviewer #1: Yes

Reviewer #2: (No Response)

6. Review Comments to the Author

**Reviewer #1: **All comments have been nicely addressed and I have nothing to add. Really nice work that is timely and will be interesting to read for others.

**Reviewer #2: **The authors have addressed some comments from this reviewer. However, there are some existing points that the authors should address them to improve this study.

1.The keywords must be implemented in this study.

2.The authors must explain why the Monte Carlo simulation method has been employed in this study.

3.The discussion part must be implemented to analyze the prioritization and the comparison with the experimental working from the proposed methods.

4.The conclusion must be re-written shortly to describe the significance of this study.

5.The quality of figures is very poor. Kindly provide the high quality solution of these figures.

7. PLOS authors have the option to publish the peer review history of their article (what does this mean?). If published, this will include your full peer review and any attached files.

Reviewer #1: No

Reviewer #2: No

---

## [Author Response · Author response to Decision Letter 1]

31 May 2024

Reviewer #1: All comments have been nicely addressed and I have nothing to add. Really nice work that is timely and will be interesting to read for others.

We want to thank the reviewer for reviewing our manuscript again, and are happy we have been able to address all comments

Reviewer #2: The authors have addressed some comments from this reviewer. However, there are some existing points that the authors should address them to improve this study.

reply: We have tried to address these remaining concerns by changing the manuscript in some specific sections:

1.The keywords must be implemented in this study.

reply: We have added the following keywords: Natural experiment, Interrupted time series, Segmented regression, Simulation study, Power. All are named numerus times in the manuscript and play a vital role in the article. 

2.The authors must explain why the Monte Carlo simulation method has been employed in this study.

reply: We have added a short section in the methods section to explain why we use a Monte Carlo simulation study: "A Monte Carlo simulation study is used to assess the performance of different ITS designs over 384 scenarios. Monte Carlo simulation allows us to estimate our statistics of interest by eliminating the noise caused by random sampling [51]. This allows us to inspect under which circumstances the ITS designs can estimate an effect well. In the simulation study, data are generated based on predefined parameters. These are then compared to the estimates to evaluate the performance of the models."

3.The discussion part must be implemented to analyze the prioritization and the comparison with the experimental working from the proposed methods.

reply: we changed the discussion section in two ways (also to adress the next comment).

1. We added a sentence to the main findings, to illustrate the implications of the simulation: "This research has investigated the performance of the standard ITS design as well as that of various extensions, in situations with limited data. While the ITS design works well under many different circumstances, there are some factors to consider before running an analysis: the sample size in the case of a step change, the number of time points for a slope change, and the problem of underfitting\\overfitting. Not one of these factors alone can account for the power, it always depends on the a combination of factors.

2. We have added a conclusion section (see also next comment)

4.The conclusion must be re-written shortly to describe the significance of this study.

reply: we have restructured the discussion section, and now added a separate conclusion section     

5.The quality of figures is very poor. Kindly provide the high quality solution of these figures. 

reply: the figures are now all available as vector-based figures, so quality issues should no longer be there.

---

## [Decision Letter · Decision Letter 2]

5 Jun 2024

The performance of interrupted time series designs with a limited number of time points: Learning losses due to school closures during the COVID-19 pandemic

PONE-D-23-39424R2

Dear Dr. van Leeuwen,

We’re pleased to inform you that your manuscript has been judged scientifically suitable for publication and will be formally accepted for publication once it meets all outstanding technical requirements.

Kind regards,

Chenfeng Xiong

Academic Editor

PLOS ONE

Additional Editor Comments (optional):

Reviewers' comments:

Reviewer's Responses to Questions

**Comments to the Author**

1. If the authors have adequately addressed your comments raised in a previous round of review and you feel that this manuscript is now acceptable for publication, you may indicate that here to bypass the “Comments to the Author” section, enter your conflict of interest statement in the “Confidential to Editor” section, and submit your "Accept" recommendation.

Reviewer #2: All comments have been addressed

2. Is the manuscript technically sound, and do the data support the conclusions?

Reviewer #2: Yes

3. Has the statistical analysis been performed appropriately and rigorously? 

Reviewer #2: Yes

4. Have the authors made all data underlying the findings in their manuscript fully available?

Reviewer #2: Yes

5. Is the manuscript presented in an intelligible fashion and written in standard English?

Reviewer #2: Yes

6. Review Comments to the Author

Reviewer #2: The authors have addressed all comments from this reviewer. The current version now looks nice and could be accepted to publish on Plos One.

7. PLOS authors have the option to publish the peer review history of their article (what does this mean?). If published, this will include your full peer review and any attached files.

Reviewer #2: No

---

## [Editor Report · Acceptance letter]

14 Jun 2024

PONE-D-23-39424R2 

PLOS ONE

Dear Dr. van Leeuwen, 

I'm pleased to inform you that your manuscript has been deemed suitable for publication in PLOS ONE. Congratulations! Your manuscript is now being handed over to our production team.

Kind regards, 

on behalf of

Dr. Chenfeng Xiong 

Academic Editor

PLOS ONE